# Regio- and chemoselective catalytic partial transfer hydrogenation of quinolines by dinuclear aluminum species

Xufang Liu, Arseni Kostenko ⓘ , Matthew M. D. Roy, Tobias Weng & Shigeyoshi Inoue ⓘ ✉

Catalytic reduction of quinolines has gained continuous interest in both academia and industry, providing direct and efficient access to tetrahydroquinolines or 1,2-dihydroquinolines. The catalytic preparation of tetrahydroquinolines has been extensively studied by transition metal complexes. By contrast, the related catalytic synthesis of 1,2-dihydroquinolines remains underdeveloped due to the difficulties in achieving precise control over both chemo- and regioselectivity. Here, we demonstrate a catalytic 1,2-reduction of quinolines using a dinuclear aluminum complex, allowing for the highly selective synthesis of a rich array of 1,2-dihydroquinolines through the combination of bimetallic catalysis and metal-ligand cooperation. These investigations showcase the capacity of main group metal catalysts to achieve reactivities distinct from well-studied transition-metal catalysis.

Catalytic reduction of N-heteroarenes is of particular interest in both organic synthesis and industrial production, offering a versatile platform for preparing saturated nitrogen-containing heterocycles[1–5]. In this context, catalytic (transfer)hydrogenation of quinolines stands out as one of the most practical transformations, and could provide the most straightforward access to tetrahydroquinolines (THQs) or 1,2-dihydroquinolines (1,2-DHQs) (Fig. 1a). (Transfer)hydrogenation of quinolines to THQs has been well-established using transition metal catalysts[6–15], and recently realized by magnesium complexes[16]. However, the related catalysis for producing 1,2-DHQs has rarely been studied. 1,2-DHQs represent a very important class of compounds, as they constitute privileged structural units in a large number of natural products and pharmacologically active therapeutic agents (Fig. 1b)[17–20]. Moreover, 1,2-DHQs serve as versatile synthetic building blocks for constructing complex organic frameworks through diverse functionalization strategies, such as olefin difunctionalization, C–H functionalization and N-functionalization (Fig. 1b)[21–24].

The lack of efficient methods for synthesizing 1,2-DHQs is likely due to a two-fold challenge associated with the control of both chemoselectivity and regioselectivity during the reduction process (Fig. 1c). First, the reaction always suffers from undesired over-

reduction of 1,2-DHQs to THQs[14,25]. Second, the inherent thermal instability of 1,2-DHQs renders their facile dehydrogenation to regenerate quinolines[26]. A major breakthrough was achieved by Wang group in 2020 using a cobalt-amido cooperative catalytic system, providing access to a broad range of 1,2-DHQs with high levels of regio- and chemoselectivity[27].

Transition-metal-based catalysts are central to modern catalytic methodologies, owing to their ability to reversibly shift between multiple oxidation states. However, their high cost and, in some cases, inherent toxicity have driven the pursuit of alternative catalytic systems. Earth-abundant main group metals have emerged as promising alternatives due to their sustainability and environmental compatibility[28–33]. Nevertheless, compared to their transition metal counterparts, main group metal catalysts remain underdeveloped, primarily due to the absence of partially filled *d* orbitals and the inherent instability of many main group metal complexes[34,35]. Aluminum, constituting approximately 8% of the Earth's crust, is the most abundant metal, making it an ideal surrogate for transition-metal catalysts[36–39]. The past decade has witnessed advancements in aluminum catalysis for reactions such as Lewis acid catalysis[39], hydroboration[40–43], hydrogenation[44–46], dehydrogenation[47], ammonia

Department of Chemistry, Institute of Silicon Chemistry and Catalysis Research Center, TUM School of Natural Sciences, Technische Universität München, Garching bei München, Germany. ✉e-mail: s.inoue@tum.de

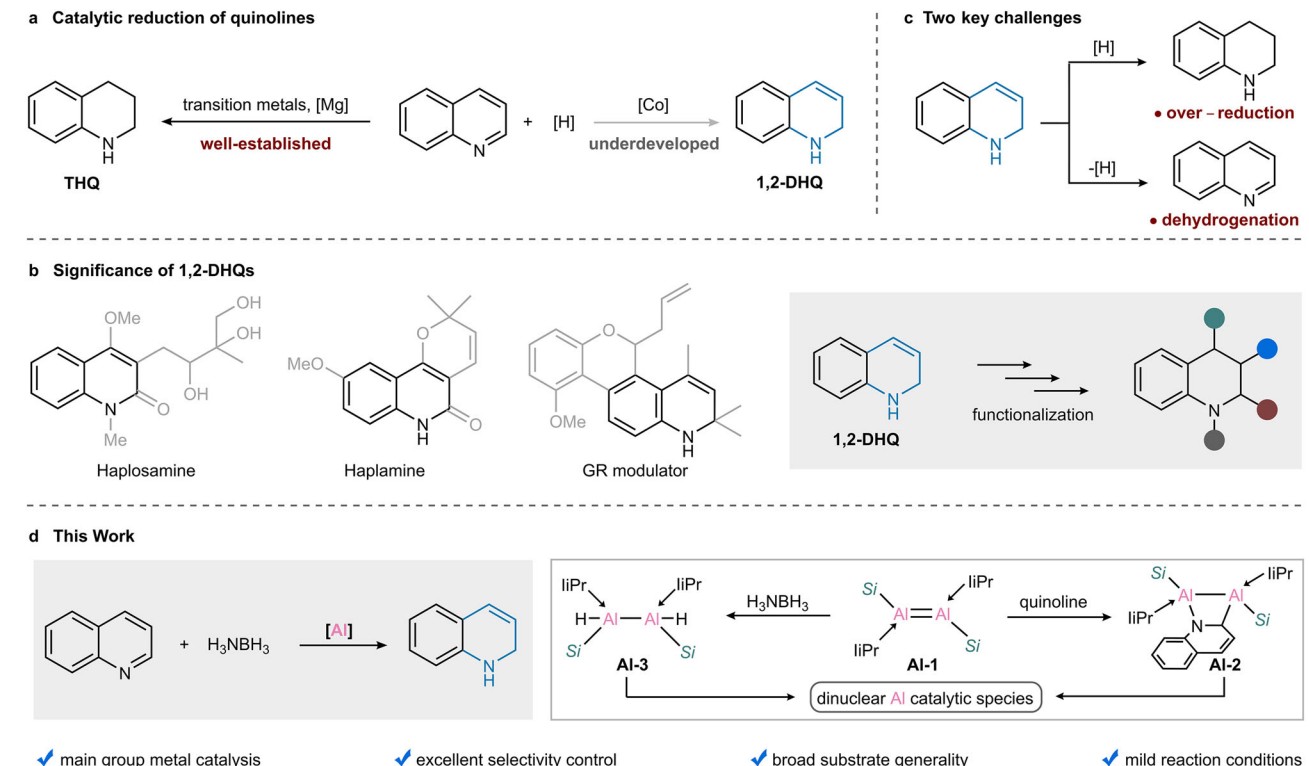

**Fig. 1 | Catalytic reduction of quinolines. a** Background for catalytic reduction of quinolines to THQs or 1,2-DHQs. **b** Significance of 1,2-DHQs. **c** Key challenges for 1,2-reduction of quinolines to 1,2-DHQs. **d** Catalytic 1,2-reduction of quinolines by a dialuminum complex.

transfer[48], and photocatalysis[49]. Nonetheless, the scope of applications remains largely confined to benchmark reactions that have been well-studied by transition-metal catalysts. Therefore, a next-stage research objective would be to uncover the unique and untapped catalytic potential of main group catalysts.

Metal−ligand cooperativity (MLC) has become a potent strategy in transition metal catalysis, which can modulate bond-breaking and bond-forming steps to enable mild and selective transformations[50,51]. However, this strategy has scarcely been utilized in main group metal catalysis[16,52−54].

Dialumenes, as dinuclear aluminum compounds, have shown great potential in bond-activating reactions and catalytic transformations since the first isolation of silyl-substituted dialumene by our group in 2017[55,56], thereby motivating their exploration in more challenging chemical processes. Here, we present an Al-catalyzed 1,2-reduction of quinolines using dialumene **Al-1** as a precatalyst. This reaction occurs under mild conditions, affording varieties of 1,2-dihydroquinolines with exceptional regio- and chemoselectivity (Fig. 1d). Mechanistic studies reveal that dialumene **Al-1** engages with both quinoline and ammonia borane to form **Al-2** and **Al-3**, respectively, providing key insights into the nature of possible catalytic species. Further theoretical studies disclose the key roles of the interplay between bimetallic sites and metal-ligand cooperativity in regulating the catalytic process. This protocol exemplifies the distinctive catalytic potential of main group metals in contrast to the well-established reactivity paradigms of transition metals. The versatility of this aluminum system is further demonstrated by the reduction of other N-heteroarenes.

## Results
### Reaction development
In line with our continuing interests in exploiting eco-friendly main group catalysts, we began our studies by using dialumene as a dinuclear aluminum catalyst for the (transfer) hydrogenation of quinoline.

After an extensive evaluation of reaction conditions (Supplementary Table 1), we were delighted to find that a combination of dialumene **Al-1** (5 mol %) as the precatalyst and $H_3NBH_3$ (1 eq.) as the reducing agent in $C_6D_6$ at room temperature delivered optimal yields of the desired 1,2-dihydroquinoline (**2a**) with 94% yield and 16:1 selectivity (Fig. 2, Equation (1)). Control experiments confirmed that the aluminum complex is responsible for the catalysis, as no reaction occurred in the absence of **Al-1**. Solvent screening revealed that non-coordinating solvents such as $C_6D_6$ and Tol-$d_8$ afforded higher selectivity, whereas the coordinating solvent THF-$d_8$ resulted in reduced selectivity (Supplementary Table 1, entries 3−5). Under optimized conditions, we further investigated the reaction outcomes using various mononuclear aluminum precatalysts (Supplementary Table 1, entries 6−11). Notably, the dialuminum species exhibited markedly superior catalytic activity compared to the mononuclear species, underscoring the distinct advantages of bimetallic catalysis in facilitating the transformation of challenging substrates. We also evaluated the catalytic performance of **Al-1** at reduced loadings and were pleased to find that considerable conversions could still be achieved, affording a turnover number (TON) of 160 and a turnover frequency (TOF) of 4.5 h$^{-1}$ (Supplementary Table 2).

The kinetic profile of the overall catalytic 1,2-reduction of **1a** was established to probe the underlying reaction pathway (Supplementary Fig. 1). The concentration of **1a** continuously decreased over time, with 1,2-dihydroquinoline **2a** emerging as the major product. Tetrahydroquinoline **3a**, as the minor product, gradually accumulated in the late stages of the transfer hydrogenation process.

### Stoichiometric experiments with dialumene
Stoichiometric experiments were then conducted to elucidate possible reaction intermediates. The reaction of dialumene **Al-1** with one equivalent of quinoline in benzene at room temperature resulted in the isolation of 1,2-dearomatized dialumina-heterocycle **Al-2** in 56% yield (Fig. 2, Equation 2). The $^1$H NMR spectrum shows four

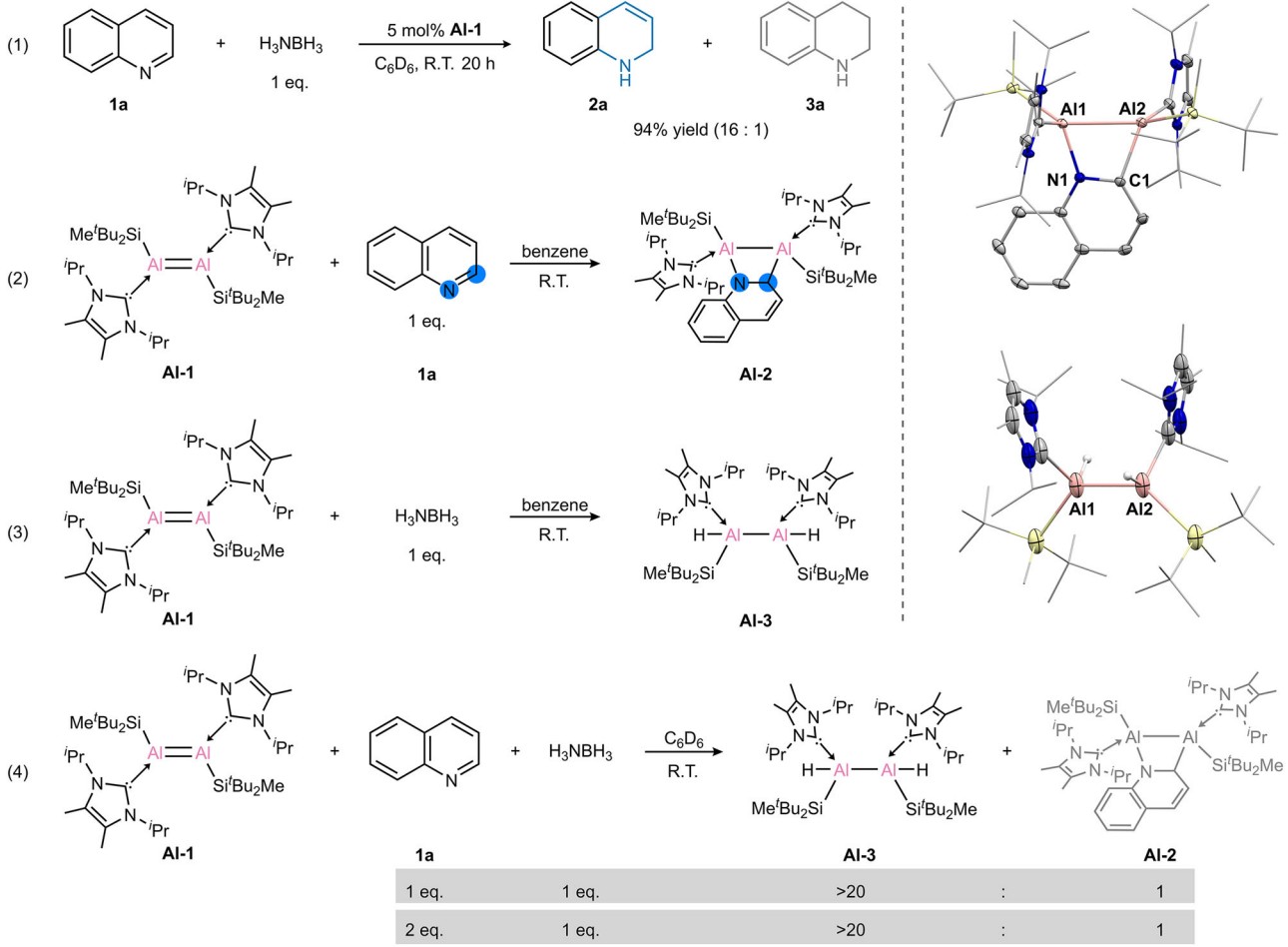

**Fig. 2 | Reaction development and stoichiometric reactivities. Al-1** as a precatalyst for the selective 1,2-reduction of quinoline and its stoichiometric reactivity with quinoline and ammonia borane.

disparate olefinic peaks in the range of 4.5–7.1 ppm, suggesting the potential dearomatization of the heterocycle. In addition, the four heptets at δ 6.23, 5.88, 4.93 and 4.69 ppm correspond to the $^i$Pr groups in NHC, revealing an unsymmetrical coordination environment around the dialumina center. The molecular structure of **Al-2** was further determined by single-crystal X-ray diffraction (SC-XRD) analysis. The core structure of **Al-2** possesses a slightly distorted four-membered Al₂CN ring, with the C1 and N1 atoms of quinoline binding to the Al2 and Al1 atoms, respectively. The loss of double bond character was confirmed by the considerable elongation of the Al1−Al2 bond length (2.6408(10) Å) compared with that of dialumene **Al-1** (2.3943(16) Å). The Al2−C1 and Al1−N1 bond lengths are 2.069(2) Å and 1.905(2) Å, respectively, typical for Al−C and Al−N single bonds.

Notably, activating C=N bonds in N-heteroarenes is a significant challenge due to the inherent difficulty in breaking the aromaticity of heterocycles, which has historically been dominated by transition metal complexes owing to their redox versatilities[57–60]. In main group chemistry, single-site Al complexes, such as NacNacAl and bis(silylene)-stabilized aluminylene, have shown distinct reactivity patterns towards quinoline, yielding the 2,2'-coupling product and the 1,4-dearomatization product, respectively[61,62]. However, the ability of dinuclear Al complexes in this field remains unexplored, offering the possibilities of unique activation modes. Thus, the isolation of **Al-2** represents a rare example of N-heteroarene activation mediated by main group metals[63–68]. This regioselective 1,2-dearomatization

reaction mode originates from the synergy between two adjacent Al$^I$ centers, which is distinct from reported mononuclear Al systems[61,62].

Building upon the observed reactivity of dialumene with quinoline to yield the dialumina-heterocycle **Al-2**, we investigated its stoichiometric reaction with H₃NBH₃. The reaction of dialumene with one equivalent of H₃NBH₃ in benzene at room temperature resulted in the isolation of **Al-3** in 41% yield (Fig. 2, Equation 3). A broad resonance at 4.43 ppm in the ¹H NMR is assigned to the hydride bound to the quadrupolar aluminum nucleus, which was further supported by the detection of a characteristic Al−H stretch at 1658 cm⁻¹ in the IR spectrum. The molecular structure of **Al-3** was unambiguously confirmed by SC-XRD analysis. The Al1−Al2 bond distance is 2.623(3) Å, typical for Al−Al single bond.

Quantum calculations were performed to elucidate the mechanisms of the reactions of dialumene with quinoline and H₃NBH₃. The calculations (Fig. 3a) show that dialumene **Al-1** is expected to react with both quinoline and ammonia borane in an exergonic fashion by 23.7 and 43.1 kcal mol⁻¹, respectively. The reaction of dialumene with quinoline is a [2 + 2] concerted cycloaddition between the Al−Al and N$^1$−C$^2$ bonds. The formation of **Al-3** also proceeds in a single step, via a cyclic six-membered transition state, by a proton and a hydride transfer from NH₃BH₃ in syn fashion. The resulting aminoborane (NH₂BH₂) oligomerizes to CBT[69]. Both reactions are irreversible, however, the barrier for the reaction of **Al-1** with quinoline, i.e. **TS1** at ΔG = 22.8 kcal mol⁻¹, is substantially higher in energy than **TS2** at ΔG = 9.0 kcal mol⁻¹, which leads to the formation of **Al-3**. Given the large disparity in energy barriers, a competitive reaction between

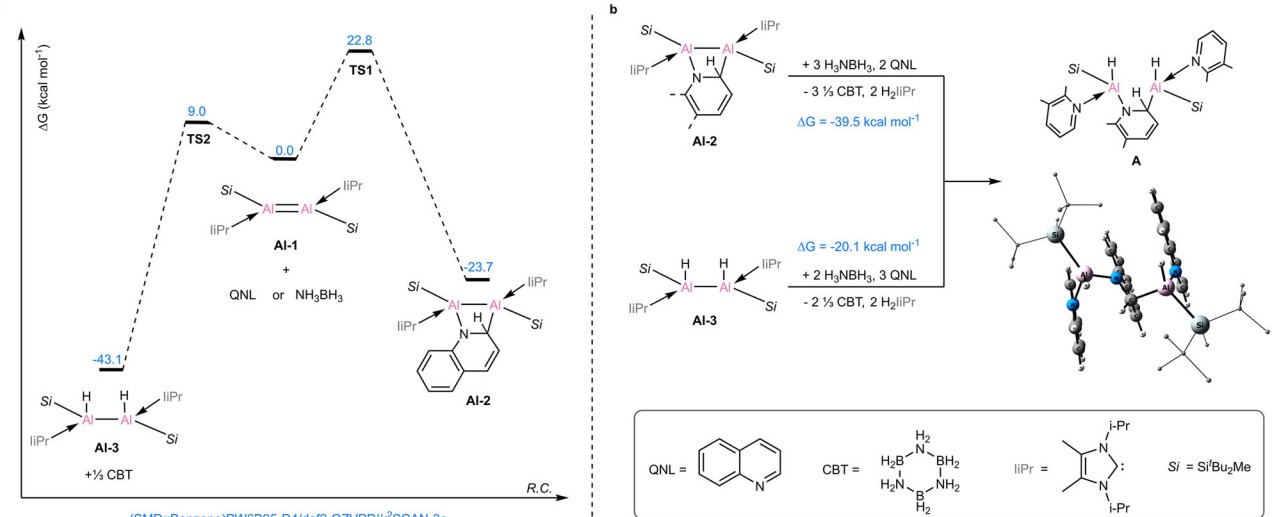

**Fig. 3 | Calculated reaction pathways for the formation of Al-2, Al-3, and A.** **a** calculated mechanism for the reaction of dialumene **Al-1** with quinoline or ammonia borane to form the respective products **Al-2** and **Al-3** (free energies at the (SMD=Benzene)PW6B95-D4/def2-QZVPP//r²SCAN-3c level of theory); **b** calculated free energy of formation of complex **A** from **Al-2** and **Al-3** in the presence of H₃NBH₃ and quinoline. In the optimized structure of **A**, the alkyl substituents are shown as wireframes, and their hydrogens are omitted for clarity.

quinoline and H₃NBH₃ was carried out (Fig. 2, Equation 4), indicating that dialumene reacts markedly faster with H₃NBH₃ than with quinoline, in line with the computational results. It should be noted that the optimized structure of **Al-3** obtained by the syn addition via **TS2** corresponds to a rotamer in which the NHCs ligands as well as the silyl substituents are nearly anti-periplanar. This is in contrast to the experimentally observed geometry **Al-3^Xray** in which the NHCs groups are in gauche orientation (Supplementary Fig. 51). The anti-periplanar conformation can be achieved upon rotation around the Al-Al bond, and the optimized structure of **Al-3^Xray** is energetically less favorable ($\Delta E = 4.5$ kcal mol⁻¹). We attribute the observation of the less favorable rotamer to crystal packing effects.

## Mechanistic investigation

As shown above, dialumene reacts with both quinoline and H₃NBH₃ to form **Al-2** and **Al-3**, featuring a heterocycle backbone and hydride substituents, respectively (Fig. 2). Then, control experiments between **Al-2/Al-3**, quinoline, and ammonia borane were conducted (Supplementary Fig. 35), aiming to uncover potential catalytic intermediates. **Al-3** is unreactive toward quinoline in the absence of H₃NBH₃, with 1,2-dihydroquinoline formed upon the addition of excess H₃NBH₃. Similarly, the reaction between **Al-2** and H₃NBH₃ does not produce any identifiable new aluminum species in the absence of quinoline, while primarily yielding 1,2-dihydroquinoline when excess quinoline is present. Both **Al-2** and **Al-3** are catalytically active toward the 1,2-reduction of quinoline (Supplementary Table 3), suggesting that the same catalytic species likely forms in both cases in the presence of excess quinoline and ammonia borane. Furthermore, the detection of hydrogenated NHC in the stoichiometric reactions involving H₃NBH₃ (Supplementary Fig. 35) suggests an NHC ligand dissociation from the aluminum coordination sphere, forming a catalytic species with an open coordination site, which is crucial for binding with an additional quinoline substrate.

To gain further insight into potential catalytic intermediates, the pyridine-bridged dialuminium complex **Al-4** was synthesized by reacting dialumene with one equivalent of pyridine (Fig. 4a). Complex **Al-4** was fully characterized by multinuclear NMR spectroscopy and X-ray diffraction. The ¹H NMR spectrum shows four disparate olefinic peaks in the range of 4.5−6.9 ppm, assigned to the dearomatized pyridine moiety. SC-XRD analysis revealed that the **Al-4** features an

Al₂CN ring similar to that of **Al-2**, with Al1−Al2, Al2−C1 and Al1−N1 bond distances of 2.609 Å, 1.957 Å and 1.969 Å, respectively (Fig. 4b). The use of **Al-4** as a precatalyst for quinoline reduction necessitated a higher temperature for reaction completion (Fig. 4c), suggesting that an N-heteroarene moiety is integrated into the catalytic species and thus influence the catalytic activity.

In light of all these observations, we propose that complex **A** (Fig. 3b), which possesses both the N-heterocycle and the hydride moieties, may serve as the catalytically active species in the reduction of quinoline. The formation of complex **A** can be achieved by the substitution of the carbene ligands with the quinoline ligand at **Al-2,** accompanied by carbene hydrogenation in the presence of ammonia borane. Similarly, **Al-3** can form complex **A** in the presence of two equivalents of ammonia borane and three equivalents of quinoline, upon release of two equivalents of CBT and two hydrogenated carbenes. Using the iMTD-GC conformational search algorithm[70] we were able to identify the low-lying conformers of **A**, and its DFT optimized structure is shown in Fig. 3b. In the dinuclear Al complex **A** the aluminum-aluminum bond is completely quenched with r(Al-Al) = 4.214 Å, and the quinoline-coordinated Al centers are situated on the opposite sides with respect to the bridging quinoline plane θ(Al-C-N-Al) = 112.3°. The nearly parallel orientation of the coordinating and the bridging quinolines is indicative of π-π stacking interactions between these moieties (Supplementary Fig. 53−54), which may facilitate the complex formation. The formation of **A** from **Al-2** and **Al-3** in the presence of ammonia borane and quinoline are calculated to be exergonic by 39.5 and 20.1 kcal mol⁻¹, respectively (Fig. 3b). Therefore, the formation of **A** from **Al-1** upon hydrogenation of both NHC moieties is exergonic by 63.2 kcal mol⁻¹. The proposed mechanism of the formation of the catalytic species **A** from **Al-1** was shown in Supplementary Fig. 52.

## Proposed mechanism

The proposed mechanism of the catalytic 1,2-reduction of quinoline by **A** is presented in Fig. 5. In the optimized structure of the quinoline bridged dialuminium complex **A**, the $C^2$ atom of the quinoline coordinating to $Al^1$ is found in close proximity (2.684 Å) to the hydrogen atom at the $C^2$ position of the quinoline bridge (the participating sites are highlighted in orange in Fig. 5). This prearrangement allows for a low barrier proton transfer from the bridging quinoline to the

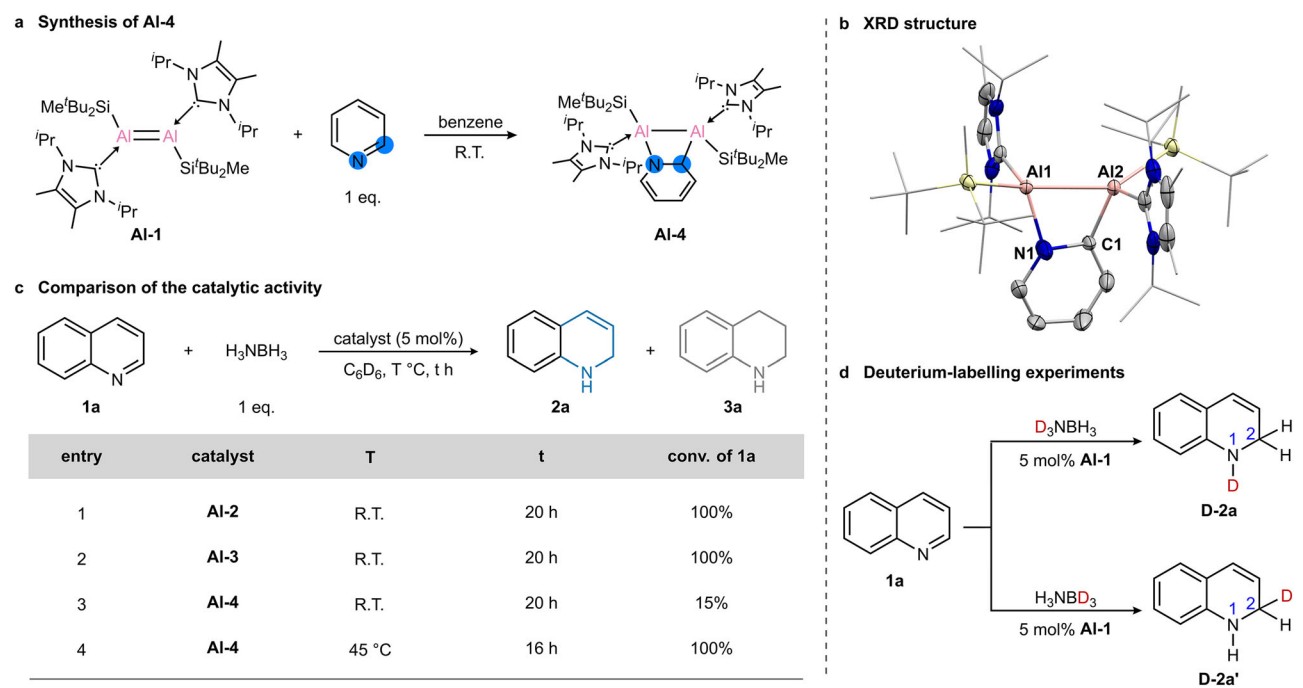

**Fig. 4 | Comparing the catalytic activities of different Al complexes and deuterium-labeling experiments. a** Synthesis of the pyridine-bridged dialuminium complex **Al-4**. **b** XRD structure of **Al-4**. **c** Comparison of the catalytic activities of **Al-2**, **Al-3** and **Al-4**. **d** Deuterium-labeling experiments using $D_3NBH_3$ and $H_3NBD_3$.

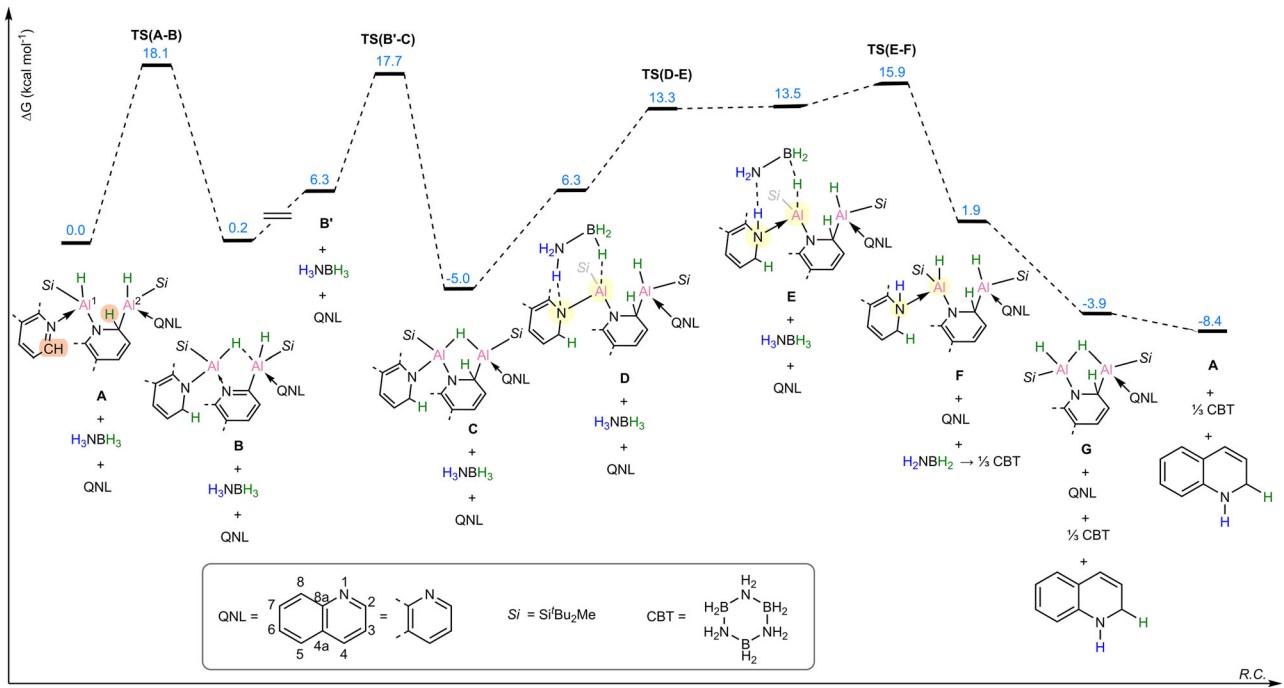

**Fig. 5 | Calculated reaction pathway for the proposed mechanism of the catalytic 1,2-reduction of quinoline.** Free energies at the (SMD=Benzene)PW6B95-D4/def2-QZVPP//r²SCAN-3c level of theory.

coordinating quinoline via a cyclic six-membered transition state **TS(A-B)** at $\Delta G = 18.1$ kcal mol⁻¹. This step is endergonic by 0.2 kcal mol⁻¹ and forms intermediate **B**. The process is accompanied by the rearomatization of the bridging quinoline and dearomatization of the external quinoline, which is now covalently bound to the Al¹ center. Thus, both aluminum centers retain the formal +3 oxidation state. Intermediate **B** isomerizes to **B'** at $\Delta G = 6.3$ kcal mol⁻¹ (this is a two-step process with a low barrier, omitted for clarity), which allows in the

following step for the hydride transfer from Al² to the C² position of the bridging quinoline. This takes place via **TS(B'-C)** $\Delta G = 17.7$ kcal mol⁻¹ forming intermediate **C** – here the bridging quinoline is again dearomatized and essentially restored to the initial state it was in **A**. In the next stage, intermediate **C** forms a complex with ammonia borane **D**, which is followed by a two-step hydrogenation of the Al¹–N fragment, via **TS(D-E)** and **TS(E-F)**. The hydride from borane is transferred to the Al center, while the proton from the ammonia is transferred to N¹ of the

external quinoline moiety. Thus, the quinoline hydrogenation is complete, and hydrogenated moiety can now be substituted by another quinoline molecule, via intermediate **G** at $\Delta G = -3.9$ kcal mol$^{-1}$, to reform the catalytic species **A** at $\Delta G = -8.4$ kcal mol$^{-1}$. According to this proposed mechanism **TS(E-F)** at $\Delta G = 15.9$ kcal mol$^{-1}$ is the rate-determining transition state since the barrier from **C** to **TS(E-F)** of 20.9 kcal mol$^{-1}$ is the highest barrier that needs to be overcome in the process.

The proposed mechanism, backed by quantum chemical calculations, relies on the distinctive properties of the quinoline-bridged dialuminium complex **A**. The specific alignment of the two N-heteroarene moieties, combined with the acidity of the hydrogen at C$^2$ of the bridging quinoline, facilitates the proton transfer to the coordinating quinoline at Al$^1$. The hydrogen atom at the C$^2$ position is then restored by the hydride shift from the Al$^2$ center. Throughout the process, the aluminum centers of the complex maintain an oxidation state of +3, which is regulated by the aromatization/dearomatization of the bridging and external N-heteroarene groups. The interplay between the active aluminum, carbon, and nitrogen centers enables the formation of complex **C**. In this complex, the hydrogenation of the Al−N bond by ammonia borane occurs. This is similar to the previously reported hydrogenation of M−N bonds by H$_3$NBH$_3$[13,27]. This step also replenishes the dialuminium complex with a hydride, which is then available in the subsequent catalytic cycle. Thus, the ability of dialumene to incorporate a bridging quinoline is a key feature in the formation of the catalytic species. During the catalytic process, the hydride sequentially migrates from ammonia borane to the aluminum centers, then to the bridging N-heteroarene, and ultimately to the coordinating N-heteroarene. Such a "relay" hydride transfer process allows for an efficient and selective transfer hydrogenation of quinoline, leveraging the synergistic effects of bimetallic centers and metal-ligand cooperativity.

The proposed mechanism (Fig. 5) can also explain the above-mentioned lower catalytic ability of complex **Al-4** (Fig. 4c). It is proposed that in order to restore the C$^2$ of the bridging quinoline, a hydride transfer from Al$^2$ to C$^2$ needs to take place, which results in the N-heteroarene moiety dearomatization. While in the case of quinoline backbone the barrier for this step **TS(B'-C)** is only 17.7 kcal mol$^{-1}$ (Fig. 5), it becomes 23.0 kcal mol$^{-1}$ when the bridging arene is a pyridine moiety (Supplementary Fig. 56). This is due to the greater challenge associated with the pyridine dearomatization compared to the heterocyclic moiety of quinoline. The reluctance of the pyridine bridge to dearomatize is also reflected in the endergonic character of this step, which makes intermediates **B** $_\text{pyridine}$ and **B'** $_\text{pyridine}$, the lowest-energy intermediates in the catalytic pathway. Subsequently, the barriers for transition state corresponding to the hydrogenation of the Al$^1$-N moiety by ammonia borane also increase (Supplementary Fig. 56). Thus, the higher barriers for the hydride transfer and the Al$^1$-N hydrogenation in the case of pyridine bridge explains the higher temperature needed for the catalytic hydrogenation of quinoline when **Al-4** is used as precatalyst.

To further substantiate the proposed mechanism, the regiospecificity of hydrogen transfer was examined using partially deuterated ammonia borane derivatives, D$_3$NBH$_3$ and H$_3$NBD$_3$ (Fig. 4d). In the reaction of **1a** with D$_3$NBH$_3$, deuterium incorporation at the 1-position was observed, leading to the clean formation of **D-2a**. Conversely, when H$_3$NBD$_3$ was used, deuterium was exclusively transferred to the 2-position, yielding **D-2a'** as the sole product. These results align with the proposed mechanism. Kinetic isotope effects (KIE) were further investigated for the catalytic reduction of quinoline (Supplementary Fig. 43). By comparing the initial rates of **2a** formation using H$_3$NBH$_3$ and D$_3$NBH$_3$, a KIE of 2.10 was determined. A slightly larger KIE of 2.43 was observed in the case of H$_3$NBD$_3$. These findings of the primary kinetic isotope effect suggest that hydrogen transfer participates in the turnover-determining step that is according to the proposed

mechanism corresponds to **TS(E-F)**. Calculations of the corresponding isotopomers show that the barrier for the catalytic system using D$_3$NBH$_3$ and H$_3$NBD$_3$ are by 0.16 kcal mol$^{-1}$ and 0.35 kcal mol$^{-1}$ higher than that for the non-deuterated system, which is consistent with the experimentally observed trend (Supplementary Fig. 57). Thus, the kinetic isotope effect experiments support the proposed rate determining transition state **TS(E-F)**. In addition, the kinetic order of each reaction component was determined for the transformation of **1a** to **2a** to gain deeper mechanistic insights. The initial rates of the catalytic reactions were measured with a series of concentrations of **Al-1**, quinoline **1a**, and ammonia borane (Supplementary Figs. 44−46). A zero-order dependency on the concentrations of quinoline **1a** was observed, suggesting that the activation of the quinoline substrate is not involved in the turnover-limiting step. The rate dependencies on **Al-1** and ammonia borane were first order, implying that the hydrogen transfer from ammonia borane to the aluminum centers is involved in the turnover-limiting step.

Based on detailed DFT calculations, a plausible catalytic cycle was proposed (Fig. 6). The catalytic cycle begins with hydrogen transfer from the bridging quinoline to the external quinoline, which generates species **B** through aromatization of the bridging quinoline and concurrent dearomatization of the external quinoline. Subsequent hydrogen migration from the aluminum center to the bridging quinoline forms species **C**, followed by hydrogenation of the Al−N bond with ammonia borane via a six-membered transition state to form **F**. Finally, ligand exchange with quinoline releases the 1,2-dihydrogenated product and regenerates the catalytic species **A**.

## Substrate scope

To investigate the generality and robustness of this catalytic system, the substrate scope was then explored (Fig. 7). Quinoline derivatives, containing both electron-donating and electron-withdrawing groups, were well tolerated. Halogen functional groups, including F, Cl, Br, were compatible with the reaction conditions. A systematic investigation of the substituent effect revealed distinct trends in reactivity and selectivity, governed by the steric and electronic properties of the substituents. Reduction of substrates with 5-, 6-, or 7-substituents yielded the corresponding 1,2-DHQs with good yields and selectivities (**2b-2d**). The solid-state structure of **2b** was unambiguously confirmed by SC-XRD analysis. Bulkier substrates with 3- or 4-substituents generally required higher temperatures, but led to enhanced selectivities (**2e-2g**). However, the reduction of 2-methylquinoline failed (**2 h**), possibly due to steric hindrance imposed by the 2-methyl substituent. Additionally, substrates bearing electron-withdrawing groups tended to exhibit lower selectivities, which may be attributed to the increased reducibility of the heterocyclic core in these cases. The applicability of this catalytic system was further investigated in the reduction of other N-heterocycles. Heterocycles, such as quinoxaline (**2i**) and benzo[d]oxazole (**2j**), were all efficiently hydrogenated to furnish the desired products with only the C=N bonds reduced. Furthermore, polycyclic N-heterocycles were also suitable substrates for this transformation, delivering the hydrogenated products with high yields and selectivities (**2k-2l**). The catalytic performance of this dinuclear aluminum system makes it as a valuable complement to existing transition metal systems, which is expected to shed light on the design and synthesis of more efficient and selective main group metal catalysts.

In summary, we report the Al-catalyzed selective 1,2-reduction of quinolines, offering a direct and efficient synthetic route to various 1,2-dihydroquinolines with excellent selectivity. In general, the presented methodology features mild operating conditions, exquisite chemo- and regioselectivity, good functional group tolerance, and broad applicability. Theoretical studies reveal the critical role of the dinuclear aluminum complex during catalysis. That is, the two pre-installed aluminum sites enable hydride transfer from ammonia borane to the

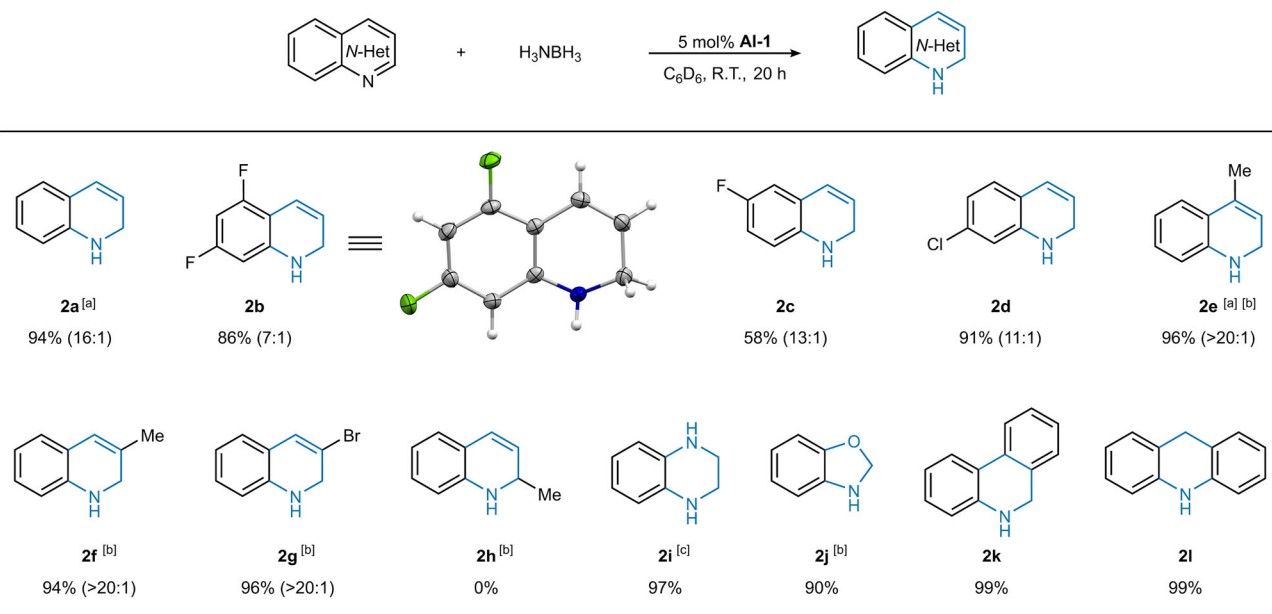

**Fig. 6 | The proposed catalytic cycle.** Proposed catalytic cycle for the selective 1,2-reduction of quinoline.

**Fig. 7 | Substrate scope.** Reactions were performed on a 0.1 mmol scale using 5 mol% [**Al-1**] as precatalyst. Unless otherwise noted, all yields refer to isolated products, and the product ratios of 1,2-DHQ and THQ (shown in parentheses) were determined by [1]H NMR analysis. [a]NMR yields are shown. [b]T = 40 °C, t = 16 h. [c]2 eq. of H₃NBH₃ was used.

quinoline substrate in a "relay" fashion, governed by the aromatization-dearomatization of the bridging and coordinating N-heteroarene motifs. This represents a rare example of Earth-abundant main-group metal catalysis in (transfer)hydrogenation reactions. And the achievement of 1,2-selectivity demonstrates the unique ability of main-group-metal catalysis to access distinctive reactivity compared to well-established transition-metal catalysis. In addition, we display the capability of dialumene in the stoichiometric activation of N-heteroarenes, including quinoline and pyridine, with exclusive selectivity for 1,2-dearomatization. This reactivity pattern contrasts with that of single-site Al complexes, highlighting the unique and pivotal role of bimetallic cooperativity in bond activation. Overall, this study demonstrates the dual role of dialuminum species in facilitating both catalytic transformation and bond activation through

bimetallic synergy and metal-ligand cooperativity, which opens avenues for the development of innovative catalytic processes and the activation of robust molecules mediated by main-group metal complexes. Further studies aimed at expanding aluminum-mediated catalytic transformations are currently underway.

## Methods

### Synthetic methods

All experiments and manipulations were carried out under an argon atmosphere using standard Schlenk or glovebox techniques. The glassware was heat-dried under vacuum prior to use. Solvents were dried by standard methods (withdrawal from MBraun Solvent Purification System and storage over molecular sieves, or distilled from sodium/benzophenone or $CaH_2$ under argon atmosphere and degassed via freeze-pump-thaw cycling). Standard chemicals were purchased from commercial suppliers and used as received if not stated otherwise.

### Spectroscopic methods

All NMR samples were prepared under argon in J. Young PTFE tubes. NMR spectra were recorded on a Bruker AV400US, DRX400, AVHD300 and AV500cr at ambient temperature if not stated otherwise. $^1H$ and $^{13}C$ NMR spectra were calibrated against the residual proton and natural abundance carbon resonances of the respective deuterated solvent as internal standard. Liquid Injection Field Desorption Ionization Mass Spectrometry (LIFDI-MS) was measured directly from an inert atmosphere glovebox with a Thermo Fisher Scientific Exactive Plus Orbitrap equipped with an ion source from Linden CMS. ATR-FTIR spectra were recorded on a PerkinElmer FTIR spectrometer (diamond ATR, Spectrum Two; located inside an argon-filled glovebox) in a range of 400–4000 $cm^{-1}$.

### Crystallographic methods

Single crystal diffraction data were collected on a single-crystal X-ray diffractometer equipped with a Charge-Integrating Pixel Array Detector (Brucker Photon-II), a Microfocus X-Ray Source with a $CuK_\alpha$ ($\lambda = 1.54178$) or a Turbo X-Ray Source rotating anode with $MoK_\alpha$ radiation ($\lambda = 0.71073$ Å) and a Helios optic using the APEX4 software package. The measurements were performed on single crystals coated with the perfluorinated ether Fomblin Y. The crystals were fixed on the top of a micro sampler, transferred to the diffractometer and frozen under a stream of cold nitrogen. Additional details for data processing, structure refinement and graphic depictions are given in the Supplementary Information.

### Computational methods

The quantum chemical calculations were conducted using ORCA 6 software[71,72]. Geometry optimizations were carried using the r²SCAN-3c composite method, utilizing the regularized and restored SCAN functional, geometrical counterpoise correction gCP, the atom-pairwise dispersion correction based on tight binding partial charges (D4), the def2-mTZVPP basis set and def2-mTZVPP/J auxiliary basis set. The optimized geometries were verified as minima or transition states by analytical frequency calculations. The transition states were additionally verified by IRC calculations. Single point calculations of the optimized geometries were conducted at the r²SCAN-3c level using the SMD solvation module, to obtain electrostatic contribution ($G_{enp}$) and the cavity term ($G_{cds}$), in order to account for the solvent effects. For more accurate electronic energies, single point calculations of the optimized geometries were carried out using the PW6B95 functional with D4 dispersion correction, def2-QZVPP basis set, def2/J and def2-QZVPP/C auxiliary basis sets. The method by which the free energies were obtained is denoted as (SMD = Benzene)PW6B95-D4/def2-QZVPP//r²SCAN-3c, and the summary of the thermochemistry results is presented in the Supplementary Information.

## Data availability

The data that support the findings of this study are available within the main text and its Supplementary Information. Crystallographic data for the structures reported in this Article have been deposited at the Cambridge Crystallographic Data Centre, under deposition numbers CCDC 2441311 (**Al-2**), 2441312 (**Al-3**) and 2441313 (**Al-4**), 2441314 (**2b**). Copies of the data can be obtained free of charge via https://www.ccdc.cam.ac.uk/structures/. All data are available from the corresponding author upon request. Source Data are provided with this manuscript. Source data are provided with this paper.

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

## Acknowledgements
This project has received funding from the Alexander von Humboldt Foundation for Research Fellowships (to X.L. and M.M.D.R.), as well as the European Union's Horizon 2020 research (ALLOWE101001591) and innovation program under the Marie Skłodowska-Curie grant agreement No. 899987. The authors gratefully acknowledge the scientific support and HPC resources provided by the Erlangen National High Performance Computing Center (NHR@FAU) of the Friedrich-Alexander-Universität Erlangen-Nürnberg (FAU) under the NHR project b255bb. NHR funding is provided by federal and Bavarian state authorities. NHR@FAU hardware is partially funded by the German Research Foundation (DFG)–440719683.

## Author contributions
X.L. and M.M.D.R. conceived and performed the synthetic experiments and analysed the data. A.K. designed and performed the theoretical analyses. T.W. performed the liquid injection field desorption ionization mass spectrometry measurements. S.I. conceived and supervised the project. X.L., A.K. and S.I. wrote the manuscript with input and critical revision from all authors.

## Funding

## Competing interests
The authors declare no competing interests.
