## [Transparent Peer review file · Nature Communications]

Regio- and Chemoselective Catalytic Partial Transfer Hydrogenation of Quinolines by Dinuclear Aluminum Species

Corresponding Author: Professor Shigeyoshi Inoue

Version 0:

Reviewer comments:

Reviewer #1

(Remarks to the Author)

The manuscript entitled "Regio- and Chemoselective Catalytic Partial Transfer Hydrogenation of Quinolines by Dinuclear Aluminum Species" by Inoue and coworkers describes the first report on the selective reduction of a quinolines via partial transfer hydrogenation utilising dinuclear aluminum species. The authors succeeded in isolating some of the possible intermediates via stoichiometric reactions and performed a DFT study to elucidate the mechanism. There is no doubt that the work has been competently done, however, I do not consider the work is novel enough for Nature Communication for mainly the following reasons: (a) There are several papers on MLC by main group compounds. Ref 16 is not the only work. See for example, J. Am. Chem. Soc. 2022, 144, 19115–19126; Dalton Trans. 2017, 46, 11192–11200; Angew. Chem., Int. Ed., 2024, 63, e202401702. (b) The selective 1,2-reduction of quinolines is reported with magnesium (ref 16), which negates the novelty of Al as the main group element doing the same job. (c) the compound with Al-Al double bond is not the active catalyst, which further reduces the novelty. Apart from this novelty ground, the following concerns need attention before publication elsewhere.

1. Did the authors performed any reaction with different stoichiometric such as 2 eq of NH_3BH_3 and 3 eq of quinoline with Al-1 to see the formation of the proposed active catalyst A.
2. In the competitive reaction of Al-1 [Fig2 (4)] if the equivalent of quinoline is increased to 2 does still the reaction stops at Al-3.
3. Also, it will be interesting to check the reaction of Al-2 with 2 eq of NH_3BH_3 and Al-3 with 3 eq of quinoline and 1 eq of NH_3BH_3 for the possible formation of the active catalyst if the authors can trap it via any of the spectroscopic technique.
4. Can authors comment on the TON and TOF of the catalyst.
5. Authors have used 1 eq of NH_3BH_3 for the catalysis wrt the substrate. It will be helpful if authors can explain how, it is being utilise for the active catalyst generation and conversion of the product upto 90 % yield.

Reviewer #2

(Remarks to the Author)

This manuscript by Inoue et. al. details the selective catalytic hydrogenation of quinolines at the 1,2-position using an Al(I) dialumene pre-catalyst. The active catalyst, the structure of which is somewhat ambiguous, is formed in situ through reaction with the substrates (quinoline and ammonia borane) and is proposed to be an Al(III) bimetallic species. The work is incredibly interesting, with both the selectivity of the catalytic reaction and the fact it is mediated by a main group element significant findings. The work that is presented is incredibly well carried out, with high quality analysis - the manuscript and supporting information are beautifully presented.

The authors have made substantial efforts elucidating the catalytic mechanism. The mechanistic/control reactions, provide insight into the reactivity of the pre-catalyst with the substrate and the products have been characterised to a high level. The computational analysis is detailed and well carried out, and the substrate scope is impressive. However, there has been a substantial assumption linking these two sections of the paper together. All the mechanistic probes were inclusive in determining the exact nature of the active catalyst, although indications as to the nature of the species were evident in the data. An active catalyst, A, has been proposed based on mechanistic investigations and the computational analysis looks credibly. However, I find this proposal quite speculative (and there are certainly other combinations of products that could be proposed) and I feel much more could be done to experimentally substantiate A.

- A-2 and A-3 do not react with either ammonia borane or quinoline respectively, with reaction only proceeding with excess substrate. The authors propose that A is formed through reaction of A-2/A-3 with excess substrate. Has the reaction between A-3 and 3+ equivalents of quinoline/ammonia borane been attempted to form A? The results of this experiment (even in situ analysis) would be incredibly valuable.
- Is the synthesis of a pyridine (or similar) base stabilised analogue of AI-1 possible? This could provide a more relevant pre-catalyst.
- Has NHC exchange been observed in other systems. In previous CO₂/HBpin catalysis the NHC has always been present in the active catalytic species.
- There is no mechanism/Ts for the formation of A. Given the importance of this species these would be valuable.

In general, I found the catalysis discussions quite sparse with limited data presented. This needs to be dramatically enhanced before publication.

- The initial introduction of the catalysis, the first paragraph, feels underwhelming given the importance of the results.
- Details of reaction optimisation and additional control reactions are limited, despite been described as 'extensive' (Table S1). This is not in line with what is expected of a new and highly selective catalytic offering. For example, NHC only control reaction would be valuable, as would a wider range of catalyst loadings and solvents.
- I note a change of selectivity with solvent and substrate – this is not commented on in the paper and could be elaborated.
- Reactions with relevant simple molecules, e.g. amine/NHC stabilised alanes, may be informative.
- Apart from the KIE data (which is only on initial rates for 2 formation) the data presented is based only on point kinetics. Full kinetic plots should be presented, at least for the representative 2, detailing consumption of starting material and formation of products.
- The authors should conduct the reactions at different concentrations to extract important information about the rate.
- The KIE plots to not intersect the origin, indicating an initial regime with a higher rate. Conducting the reaction a lower concentration should slow down the reaction, I am not convinced the authors are capturing the initial rates in the data presented.

Minor comments:

- No figures of transition states – please include
- Data presented (e.g. NMRs) for control reactions, not just of isolated products.

Overall, I feel this is an important and significant contribution to the field and with appropriate revisions would be suitable for publication in Nature Communications.

Reviewer #3

(Remarks to the Author)

This manuscript presents a regioselective and chemoselective method for the 1,2-reduction of quinoline, catalyzed by a dinuclear aluminum complex (AI-1) utilizing amine borane as a hydrogen source under mild conditions. The authors successfully isolated and characterized two key catalytically active intermediates (AI-2 and AI-3). A plausible catalytic mechanism was proposed based on detailed control experiments and DFT calculations, with kinetic isotope effect (KIE) experiments and substrate scope investigations further substantiating certain mechanistic hypotheses. Notably, the mechanistic studies revealed that bimetallic cooperation plays a critical role in the catalytic process.

This work explores the application of an earth-abundant main-group metal catalyst for the partial reduction of N-heteroarenes, contributing to the broader development of sustainable catalytic strategies based on non-transition-metal systems. Therefore, I believe this paper merits publication, pending the following minor suggestions:

1. The authors are encouraged to attempt the detection of the proposed key catalytic intermediate (Complex A) through in situ NMR monitoring or HRMS under catalytic or stoichiometric conditions, as this critical species has not yet been experimentally observed.
2. On pages 3 and 4 of the manuscript, the authors have misspelled CTB as CBT. This needs correction to ensure consistency and accuracy in terminology.
3. A systematic evaluation of the catalytic activity of the dinuclear aluminum complexes (AI-1, AI-2, and AI-3) through reduced catalyst loading experiments would be valuable, particularly in quantifying turnover numbers (TON) and turnover frequencies (TOF).
4. The inclusion of a catalytic cycle diagram in the manuscript is recommended to graphically elucidate the mechanistic pathway. This visual representation would significantly enhance conceptual clarity for readers assessing the proposed reaction mechanism.
5. The substrate scope results indicate that while 2-methylquinoline did not react, product 2I was obtained in 99% yield. The observed inactivity attributed to steric effects requires further validation. Testing 2,4-disubstituted quinolines and supporting the findings with comparative DFT calculations would be beneficial.
6. To further validate the existence of π - π stacking interactions and their significance in stabilizing intermediate A, non-

covalent interaction (NCI) plots or energy decomposition analysis (EDA) might be helpful.

Version 1:

Reviewer comments:

Reviewer #1

(Remarks to the Author)

The authors have addressed the main concerns and improved the manuscript based on the suggestions. It may be accepted for Nature Communications

Reviewer #2

(Remarks to the Author)

I am satisfied that the authors have made significant efforts to improve the manuscript based on feedback from all three reviewers. It is a shame that the proposed active catalyst A could not be detected – though this is perhaps not surprising in a very active system.

I have two minor comments, but I don't feel this hinders publication.

1. The results of the more detailed catalytic investigations are interesting. I do feel that these results could be slightly better integrated in terms of the narrative, aiding the understanding of the system.

2. The proposed formation of A from AI-1/AI-3 is quite longwinded – I would encourage the author to continue investigating the solution behaviour of AI-3 to provide experimental support for the formation of A.

I noted a few minor typos and odd abbreviations, but these can be addressed in the edit.

Otherwise, I would consider this revised manuscript of suitable standard and impact for publication on Nature Communications.

Reviewer #3

(Remarks to the Author)

The authors have taken all comments seriously and prepared a rigorous revised version of the manuscript. I am happy to recommend acceptance in its current form.

Reviewer #1 (Remarks to the Author):

The manuscript entitled “Regio- and Chemoselective Catalytic Partial Transfer Hydrogenation of Quinolines by Dinuclear Aluminum Species” by Inoue and coworkers describes the first report on the selective reduction of a quinolines via partial transfer hydrogenation utilising dinuclear aluminum species. The authors succeeded in isolating some of the possible intermediates via stoichiometric reactions and performed a DFT study to elucidate the mechanism. There is no doubt that the work has been competently done, however, I do not consider the work is novel enough for Nature Communication for mainly the following reasons: (a) There are several papers on MLC by main group compounds. Ref 16 is not the only work. See for example, J. Am. Chem. Soc. 2022, 144, 19115–19126; Dalton Trans. 2017, 46, 11192–11200; Angew. Chem., Int. Ed., 2024, 63, e202401702. (b) The selective 1,2-reduction of quinolines is reported with magnesium (ref 16), which negates the novelty of Al as the main group element doing the same job. (c) the compound with Al-Al double bond is not the active catalyst, which further reduces the novelty. Apart from this novelty ground, the following concerns need attention before publication elsewhere.

We sincerely thank the reviewer for their thorough evaluation of our manuscript and for the constructive comments that have guided our revision. However, in response to the reviewer’s concerns regarding the novelty of this work, we would like to offer the following clarification.

Regarding point (a), we thank the reviewer for the insightful comment. The relevant references have been appropriately cited in the revised manuscript. While metal–ligand cooperativity (MLC) has been explored in several examples of main-group catalysis, its application within the context of bimetallic systems remains elusive. In the revised version, we have conducted a comparative study of the catalytic performance of our dialuminum complex versus various mononuclear aluminum precatalysts. Strikingly, the dialuminum species demonstrated significantly enhanced catalytic activity, highlighting the unique advantages of bimetallic cooperativity in promoting the transformation of challenging substrates.

Regarding point (b), thanks for the comment from the reviewer. The magnesium complex is the only reported main group catalyst that can hydrogenate quinolines, however, it results in the fully hydrogenated THQ products, which contrasts with our dialuminum system exhibiting a unique 1,2-selectivity to form 1,2-DHQs. In fact, the 1,2-selective hydrogenation of quinolines has been rarely studied, despite the value of 1,2-DHQs as key synthons for complex organic frameworks, pharmaceuticals, and natural products. The lack of efficient methods is likely due to a dual challenge associated with the control of both chemoselectivity and regioselectivity. To our knowledge, precedent is only limited to a cobalt system, while the use of main group catalysts in this field has never been documented. Therefore, this study represents the first example of main-group-metal-based catalysis for the challenging 1,2-hydrogenation of quinolines.

Regarding point (c), we agree with the reviewer that dialumene is precatalyst for the catalytic transformation rather than the active catalyst, due to the difficulty of aluminum compounds to undergo reductive elimination. Nevertheless, mechanistic investigations, including stoichiometric

experiments and DFT calculations, were conducted to elucidate the potential catalytic intermediates and the underlying reaction mechanism.

In addition, the reviewer may neglect the capacity of dialumene to participate in the stoichiometric activation of heteroarenes, including quinoline and pyridine. The reactivity pattern differs from that of single-site aluminum complexes (ref 61-62), demonstrating the unique role of bimetallic cooperativity in bond activation. Considering the inherent difficulty of disrupting aromaticity in heterocycles and the limited use of main-group metals in this area, the 1,2-dearomatization of N-heteroarenes by dialumene represents a rare and noteworthy example of main-group-metal-mediated activation of inert small molecules.

Furthermore, based on the reviewers' constructive feedback, the manuscript has been revised and improved in the following aspects:

- (1) The significance of 1,2-DHQs and the challenges associated with their synthesis have been thoroughly discussed in the introduction part, further highlighting the importance and complexity of achieving 1,2-selective reduction of quinolines.
- (2) We also evaluated the catalytic performance of various mononuclear aluminum precatalysts. Notably, the dialuminum species **Al-1** exhibited markedly superior catalytic activity compared to the mononuclear species, underscoring the distinct advantages of bimetallic catalysis in facilitating the transformation of challenging substrates. Further details are provided in our responses to question 5 from Reviewer 2.
- (3) Comprehensive kinetic studies have been carried out, providing deeper mechanistic insights into the catalytic process. Further details are provided in our responses to questions 8–10 from Reviewer 2.

Overall, this work not only demonstrates the potential of the dialuminum catalyst to exhibit a distinct reactivity pattern compared to well-established transition-metal catalysis, but also showcases the unique capability of dialumene to stoichiometrically activate challenging N-heteroarenes through bimetallic cooperativity. Given the importance and inherent challenges of achieving 1,2-reduction of quinolines, the current absence of main-group catalysts capable of such selective transformations, and the demonstrated role of bimetallic cooperativity in both catalytic hydrogenation and bond activation, we believe the novelty and significance of the revised manuscript are well aligned with the standards of *Nature Communications*.

1. Did the authors performed any reaction with different stoichiometric such as 2 eq of NH_3BH_3 and 3 eq of quinoline with Al-1 to see the formation of the proposed active catalyst A.

Response: We greatly appreciate the constructive comments and suggestions from the reviewer. We tested the reaction of **Al-1** with 2 eq. of ammonia borane and 3 eq. of quinoline. Unfortunately, the proposed active catalyst A was not detected by either in-situ NMR or HRMS analysis (Figure X1). Instead, 1,2-DHQ and the hydrogenated NHC were observed by ^1H NMR. This is likely because the catalytic species is expected to form only in the presence of excess quinoline and ammonia borane; however, once formed, it may rapidly catalyze the hydrogenation of quinoline to produce 1,2-DHQ, making its direct observation challenging.

Figure X1. ^1H NMR of the reaction of **Al-1** with 2 eq. of ammonia borane and 3 eq. of quinoline.

2. In the competitive reaction of **Al-1** [Fig2 (4)] if the equivalent of quinoline is increased to 2 does still the reaction stops at **Al-3**.

Response: We thank the reviewer for this valuable suggestion. As shown in Figure X2, the competitive reaction of **Al-1** with 2 eq. of quinoline and 1 eq. of NH_3BH_3 still exclusively yielded **Al-3**. This result has been included into the revised manuscript and further supports the calculated mechanism which predicts a significantly lower energy barrier for the formation of **Al-3** compared to **Al-2**.

Figure X2. Competitive reaction of **Al-1** with quinoline and ammonia borane.

3. Also, it will be interesting to check the reaction of AI-2 with 2 eq of NH₃BH₃ and AI-3 with 3 eq of quinoline and 1 eq of NH₃BH₃ for the possible formation of the active catalyst if the authors can trap it via any of the spectroscopic technique.

Response: The reaction of **AI-2** with 2 eq of NH₃BH₃ resulted in the formation of hydrogenated carbene, however, the resulting aluminum species cannot be identified. The reaction of **AI-3** with 3 eq of quinoline and 1 eq of NH₃BH₃ gave a result similar to that described in Question 1.

4. Can authors comment on the TON and TOF of the catalyst.

Response: Thanks for the suggestion from the reviewer. As suggested, we explored the catalytic performance of **AI-1** at reduced loadings and were pleased to find that considerable conversions could still be achieved, affording a turnover number (TON) of 160 and a turnover frequency (TOF) of 4.5 h⁻¹ (Table S2).

Table S2. TON and TOF examination of **AI-1**

Reaction scheme: Quinoline (1a) + H₃NBH₃ $\xrightarrow[\text{C}_6\text{D}_6, \text{R.T., t h}]{\text{AI-1 (x mol\%)}}$ 2a + 3a

entry	x mol%	t (h)	yield of 2a	2a : 3a	TON	TOF (h ⁻¹)
1	5 mol%	20 h	94%	16 : 1	19	1 h ⁻¹
2	2 mol%	20 h	94%	>20 : 1	47	2.4 h ⁻¹
3	1 mol%	20 h	89%	20 : 1	89	4.5 h⁻¹
4	0.5 mol%	48 h	80%	16 : 1	160	3.3 h ⁻¹

5. Authors have used 1 eq of NH₃BH₃ for the catalysis wrt the substrate. It will be helpful if authors can explain how, it is being utilise for the active catalyst generation and conversion of the product up to 90 % yield.

Response: Thank you to the reviewer for this thoughtful question. In theory, NH₃BH₃ can release up to three equivalents of H₂, accompanied by the formation of various dehydrocoupling products. In our system, a series of boron-containing species were observed during the catalytic process, as confirmed by ¹¹B NMR spectroscopy (Figure X3). Therefore, one equivalent of NH₃BH₃ should be sufficient to activate the catalyst and enable substrate conversion.

Figure X3. ^{11}B NMR of the catalytic reaction.

Reviewer #2 (Remarks to the Author):

This manuscript by Inuo et. al. details the selective catalytic hydrogenation of quinolines at the 1,2-position using an Al(I) dialumene pre-catalyst. The active catalyst, the structure of which is somewhat ambiguous, is formed in situ through reaction with the substrates (quinoline and ammonia borane) and is proposed to be an Al(III) bimetallic species. The work is incredibly interesting, with both the selectivity of the catalytic reaction and the fact it is mediated by a main group element significant findings. The work that is presented is incredibly well carried out, with high quality analysis - the manuscript and supporting information are beautifully presented.

The authors have made substantial efforts elucidating the catalytic mechanism. The mechanistic/control reactions, provide insight into the reactivity of the pre-catalyst with the substrate and the products have been characterised to a high level. The computational analysis is detailed and well carried out, and the substrate scope is impressive. However, there has been a substantial assumption linking these two sections of the paper together. All the mechanistic probes were inclusive in determining the exact nature of the active catalyst, although indications as to the nature of the species were evident in the data. An active catalyst, A, has been proposed based on mechanistic investigations and the computational analysis looks credible. However, I

find this proposal quite speculative (and there are certainly other combinations of products that could be proposed) and I feel much more could be done to experimentally substantiate A.

1. Al-2 and Al-3 do not react with either ammonia borane or quinoline respectively, with reaction only proceeding with excess substrate. The authors propose that A is formed through reaction of Al-2/Al-3 with excess substrate. Has the reaction between Al-3 and 3+ equivalents of quinoline/ammonia borane been attempted to form A? The results of this experiment (even in situ analysis) would be incredibly valuable.

Response: We greatly appreciate the constructive comments and suggestions from the reviewer. We tested the reaction of **Al-3** with 3 eq. of quinoline and 3 eq. of ammonia borane. Unfortunately, the proposed catalytic species A was not detected by either in-situ NMR or HRMS analysis. Instead, ^1H NMR revealed the presence of 1,2-DHQ and the hydrogenated NHC. This likely reflects the transient nature of species A, which is expected to form only in the presence of excess quinoline and ammonia borane. Once generated, it may rapidly catalyze the hydrogenation of quinoline to 1,2-DHQ, thereby hindering its direct observation.

Figure X4. ^1H NMR of the reaction of **Al-3** with 3 eq. of ammonia borane and 3 eq. of quinoline.

2. Is the synthesis of a pyridine (or similar) base stabilised analogue of Al-1 possible? This could provide a more relevant pre-catalyst.

Response: Thank you for the reviewer's insightful suggestion. The pyridine-stabilized analogue of **Al-1** would indeed serve as a more relevant pre-catalyst and could offer indirect evidence for the proposed intermediate **A**. As recommended, we successfully synthesized the DMAP-stabilized $\text{AlI}_2(\text{Si}^t\text{Bu}_2\text{Me})$ precursor using a two-step procedure, as shown below. However, the subsequent reduction of $\text{DMAP} \rightarrow \text{AlI}_2(\text{Si}^t\text{Bu}_2\text{Me})$ resulted in an ill-defined mixture, likely due to the relatively weaker electron-donating ability of DMAP, which may be insufficient to stabilize the electron-deficient aluminum center.

Figure X5. ^1H NMR of $\text{DMAP} \rightarrow \text{AlI}_3$.

Figure X6. ^1H NMR of $\text{DMAP} \rightarrow \text{AlI}_2(\text{Si}^t\text{Bu}_2\text{Me})$.

3. Has NHC exchange been observed in other systems. In previous CO_2/HBpin catalysis the NHC has always been present in the active catalytic species.

Response: Thank you to the reviewer for this thoughtful question. In our previous work, we proposed an NHC dissociation and re-coordination mechanism for dialumene-mediated C–F activation in monofluorobenzene, wherein NHC dissociation facilitates the subsequent fluoride migration step (Ref: *J. Am. Chem. Soc.* 2024, 146, 23591). In the present study, the detection of the hydrogenated NHC in stoichiometric reactions involving ammonia borane provides compelling evidence for NHC release during the catalytic process.

- There is no mechanism/TSS for the formation of A. Given the importance of this species these would be valuable.

Response:

Figure S48. The proposed mechanism of the formation of the catalytic species **A**. The free energies (at (SMD=Benzene)PW6B95-D4/def2-QZVPP//r²SCAN-3c level of theory) relative to the starting compound **AI-1** are shown in brackets.

We propose that the active catalytic species is formed in the reaction via the pathway described in Figure S48. First, dialumene **AI-1** reacts with the ammonia borane complex to form the **AI-3**. This process is expected to proceed faster than the formation of **AI-2**, as described in the main text. Next one of the NHC ligands is replaced by a quinoline forming intermediate **I** via intermediate **H**. At this stage the free carbene is hydrogenated by an additional equivalent of the ammonia borane complex. **I** converts to intermediate **J**, achieving the addition of the quinoline across the Al-Al bond, similarly to the formation of **AI-2**. Compound **J** rearranges to **K** in which there is a coordination site for an additional quinoline molecule. Upon coordination intermediate **L** is formed. **L** can undergo the NHC – quinolone exchange accompanied by NHC hydrogenation. Thus, first NHC dissociates forming intermediate **M**, which upon binding to a quinoline forms the proposed catalytic species **A**. The free carbene is then hydrogenated in the presence of the ammonia borane complex. The figure and the explanation have been incorporated into the revised Supporting Information.

In general, I found the catalysis discussions quite sparse with limited data presented. This needs to be dramatically enhanced before publication.

4. The initial introduction of the catalysis, the first paragraph, feels underwhelming given the importance of the results.

Response: We agree with the reviewer that the initial discussion on catalysis lacked sufficient details given the significance and challenges of the catalytic 1,2-reduction of quinolines.

In the revised manuscript, we have expanded the introduction to better emphasize the importance of 1,2-dihydroquinolines (1,2-DHQs), highlighting their role as privileged structural motifs in numerous natural products and pharmacologically active compounds, as well as their value as versatile synthetic intermediates for constructing complex organic frameworks through diverse functionalization strategies.

Moreover, as suggested by the reviewer, we have broadened the catalysis section in several key aspects: (1) detailed evaluations of reaction parameters were conducted to identify the critical factors influencing the reaction; (2) the turnover number (TON) and turnover frequency (TOF) of the aluminum catalyst were assessed to evaluate its catalytic efficiency; and (3) the kinetic profile of the catalytic process was established to gain deeper mechanistic insights. Additional details can be found in our responses to questions 5–8.

5. Details of reaction optimisation and additional control reactions are limited, despite been described as ‘extensive’ (Table S1). This is not in line with what is expected of a new and highly selective catalytic offering. For example, NHC only control reaction would be valuable, as would a wider range of catalyst loadings and solvents.

Response: We thank the reviewer for this valuable suggestion, which would help showcase the high reactivity and selectivity of the dialuminum catalytic system. As suggested, we screened several solvents and evaluated the performance of a series of aluminum precatalysts. The results are summarized in Table S1.

Solvent screening revealed that non-coordinating solvents such as C_6D_6 and $Tol-d_8$ afforded higher selectivity, whereas the weakly coordinating solvent THF- d_8 resulted in reduced selectivity (Table S1, entries 3-5). However, the origin of the observed solvent effects remains unclear. One possible explanation is that coordinating solvents may bind to the metal centers, generating a more reactive but less selective catalytic species, whereas non-coordinating solvents do not interfere in this manner.

Table S1. Optimization of reaction conditions for the transfer-hydrogenation of **1a**.

entry	catalyst	[H]	solvent	yield of 2a	2a : 3a
1	none	H ₃ NBH ₃ (2 eq.)	C ₆ D ₆	< 5%	–
2	Al-1	Me ₂ HNBH ₃ (2 eq.)	C ₆ D ₆	< 5%	–
3	Al-1	H ₃ NBH ₃ (2 eq.)	C ₆ D ₆	86%	6 : 1
4	Al-1	H ₃ NBH ₃ (2 eq.)	Tol-d ₈	62%	8 : 1
5	Al-1	H ₃ NBH ₃ (2 eq.)	THF-d ₈	75%	3 : 1
6	Al-1	H₃NBH₃ (1 eq.)	C₆D₆	94%	16 : 1
7	AlCl₃	H ₃ NBH ₃ (1 eq.)	C ₆ D ₆	< 5%	–
8	Al^tBu₃	H ₃ NBH ₃ (1 eq.)	C ₆ D ₆	9%	>20 : 1
9	Al(Si^tBu₂Me)₃	H ₃ NBH ₃ (1 eq.)	C ₆ D ₆	15%	>20 : 1
10	NHC•AlH₃	H ₃ NBH ₃ (1 eq.)	C ₆ D ₆	66%	8 : 1
11	NMe₃•AlH₃	H ₃ NBH ₃ (1 eq.)	C ₆ D ₆	9%	>20 : 1

Under optimized conditions, we further investigated the reaction outcomes using various mononuclear aluminum precatalysts (Table S1, entries 6-11). Notably, the dialuminum species exhibited markedly superior catalytic activity compared to the mononuclear precatalysts, underscoring the distinct advantages of dinuclear catalysts in facilitating the transformation of challenging substrates.

We also explored the catalytic performance of **Al-1** at reduced loadings and were pleased to find that considerable conversions could still be achieved, affording a turnover number (TON) of 160 and a turnover frequency (TOF) of 4.5 h⁻¹ (Table S2).

Table S2. TON and TOF examination of **Al-1**.

entry	x mol%	t (h)	yield of 2a	2a : 3a	TON	TOF (h ⁻¹)
1	5 mol%	20 h	94%	16 : 1	19	1 h ⁻¹
2	2 mol%	20 h	94%	>20 : 1	47	2.4 h ⁻¹
3	1 mol%	20 h	89%	20 : 1	89	4.5 h⁻¹
4	0.5 mol%	48 h	80%	16 : 1	160	3.3 h ⁻¹

6. I note a change of selectivity with solvent and substrate – this is not commented on in the paper and could be elaborated.

Response: Thanks for the suggestion from the reviewer. Regarding solvent effect, it was found that non-coordinating solvents such as C₆D₆ and Tol-d₈ afforded higher selectivity, whereas the weakly coordinating solvent THF-d₈ resulted in reduced selectivity. This point has also been addressed in the response to question 5.

Regarding substrate effects, the observed selectivity appears to be influenced by both the steric and electronic properties of the substituents. Substrates with 5-, 6-, or 7-substituents yielded the corresponding 1,2-DHQs with good yields and selectivities (**2b-2d**). Bulkier substrates with 3- or 4-substituents generally led to enhanced selectivities (**2e-2g**). However, the reduction of 2-methylquinoline failed (**2h**), due to steric hindrance imposed by the 2-methyl substituent. Additionally, substrates bearing electron-withdrawing groups tended to exhibit lower selectivities, which may be attributed to the increased reducibility of the heterocyclic core in these cases.

7. Reactions with relevant simple molecules, e.g. amine/NHC stabilised alanes, may be informative.

Response: The reviewer provided a great perspective to highlight the uniqueness of bimetallic catalysis. Under optimized conditions, we evaluated the catalytic performance of various mononuclear aluminum precatalysts (Table S1, entries 6-11). Notably, the dialuminum species exhibited markedly superior catalytic activity compared to the mononuclear precatalysts, underscoring the distinct advantages of dinuclear catalysts in facilitating the transformation of challenging substrates.

8. Apart from the KIE data (which is only on initial rates for 2 formation) the data presented is based only on point kinetics. Full kinetic plots should be presented, at least for the representative 2, detailing consumption of starting material and formation of products.

Response: Thanks for the reviewer's valuable suggestion, which has helped us gain deeper mechanistic insights into the catalytic process. The reaction profile of aluminum-catalyzed 1,2-reduction of **1a** is outlined in Figure S1. The concentration of **1a** continuously decreased over time, with 1,2-dihydroquinoline **2a** emerging as the major product. Tetrahydroquinoline **3a**, as the minor product, gradually accumulated in the late stages of the transfer hydrogenation process. These results have been incorporated in the revised manuscript and supporting information.

Figure S1. Kinetic profile for the catalytic 1,2-reduction of **1a**.

9. The authors should conduct the reactions at different concentrations to extract important information about the rate.

Response: As suggested by the reviewer, the kinetic order of each reaction component was established for the transformation of **1a** to **2a** at room temperature. The initial rates of the catalytic reactions were measured with a series of concentrations of **Al-1**, quinoline **1a**, and ammonia borane (Figure S44-46). A zero-order dependency on the concentrations of quinoline **1a** was observed, suggesting that the activation of the quinoline substrate is not involved in the turnover-limiting step. The rate dependencies on **Al-1** and ammonia borane were first order, which implies that the hydrogen transfer from ammonia borane to the aluminum centers is involved in the turnover-limiting step. These results have been incorporated in the revised manuscript and supporting information.

Figure S44-46. Determination of reaction order toward each reaction component.

10. The KIE plots do not intersect the origin, indicating an initial regime with a higher rate. Conducting the reaction at a lower concentration should slow down the reaction, I am not convinced the authors are capturing the initial rates in the data presented.

Response: We thank the reviewer for pointing this out. We repeated the reaction at a lower concentration and remeasured the KIE. The updated KIE plots now nearly intersect at the origin, which should give more accurate initial reaction rates. As shown in Figure S43, a KIE of 2.10 was obtained in the case of D₃NBH₃, and a KIE of 2.43 was observed in the case of H₃NBD₃.

Figure S43. KIE for the catalytic 1,2-reduction of 1a.

Minor comments:

11. No figures of transition states – please include

Response: We made a figure with transition state drawings. It will be included in the SI. Additionally, all of the geometries are available in xyz format for convenient visualization.

12. Data presented (e.g. NMRs) for control reactions, not just of isolated products.

Response: We thank the reviewer for pointing this out. The NMR spectra for control reactions have been added to the revised supporting information.

Overall, I feel this is an important and significant contribution to the field and with appropriate revisions would be suitable for publication in Nature Communications.

Reviewer #3 (Remarks to the Author):

This manuscript presents a regioselective and chemoselective method for the 1,2-reduction of quinoline, catalyzed by a dinuclear aluminum complex (AI-1) utilizing amine borane as a hydrogen source under mild conditions. The authors successfully isolated and characterized two key catalytically active intermediates (AI-2 and AI-3). A plausible catalytic mechanism was proposed based on detailed control experiments and DFT calculations, with kinetic isotope effect (KIE) experiments and substrate scope investigations further substantiating certain mechanistic hypotheses. Notably, the mechanistic studies revealed that bimetallic cooperation plays a critical role in the catalytic process.

This work explores the application of an earth-abundant main-group metal catalyst for the partial reduction of N-heteroarenes, contributing to the broader development of sustainable catalytic strategies based on non-transition-metal systems. Therefore, I believe this paper merits publication, pending the following minor suggestions:

1. The authors are encouraged to attempt the detection of the proposed key catalytic intermediate (Complex A) through in situ NMR monitoring or HRMS under catalytic or stoichiometric conditions, as this critical species has not yet been experimentally observed.

Response: We sincerely appreciate the reviewer's valuable comments and suggestions. As suggested, we conducted reactions using **AI-1** with 3 eq. of quinoline and 3 eq. of ammonia borane; **AI-2** with 2 eq. of quinoline and 3 eq. of ammonia borane; and **AI-3** with 3 eq. of quinoline and 2 eq. of ammonia borane. Unfortunately, in all cases, the formation of the proposed catalytic species A could not be detected, either by in situ NMR or HRMS analysis. Instead, ¹H NMR analysis revealed the formation of 1,2-DHQ and the hydrogenated NHC. This is likely because the catalytic species forms only under conditions with excess quinoline and ammonia borane, but once generated, it rapidly facilitates the hydrogenation of quinoline to 1,2-DHQ, making its direct detection difficult.

2. On pages 3 and 4 of the manuscript, the authors have misspelled CTB as CBT. This needs correction to ensure consistency and accuracy in terminology.

Response: We thank the reviewer for pointing this out. This mistake has been corrected in the revised manuscript.

3. A systematic evaluation of the catalytic activity of the dinuclear aluminum complexes (AI-1, AI-2, and AI-3) through reduced catalyst loading experiments would be valuable, particularly in quantifying turnover numbers (TON) and turnover frequencies (TOF).

Response: As suggested by the reviewer, we investigated the catalytic performance of **AI-1** at reduced loadings and were pleased to find that substantial conversions were still achieved, yielding a turnover number (TON) of 160 and a turnover frequency (TOF) of 4.5 h⁻¹ (Table S2).

Table S2. TON and TOF examination of **AI-1**.

entry	x mol%	t (h)	yield of 2a	2a : 3a	TON	TOF (h ⁻¹)
1	5 mol%	20 h	94%	16 : 1	19	1 h ⁻¹
2	2 mol%	20 h	94%	>20 : 1	47	2.4 h ⁻¹
3	1 mol%	20 h	89%	20 : 1	89	4.5 h ⁻¹
4	0.5 mol%	48 h	80%	16 : 1	160	3.3 h ⁻¹

Additionally, we assessed the performance of **AI-2** and **AI-3** under similar conditions. Complex **AI-2** exhibited a TON of 45 and a TOF of 2.3 h⁻¹, while complex **AI-3** afforded a TON of 178 and a TOF of 4.4 h⁻¹ (Table S3).

Table S3. TON and TOF examination of **AI-2** and **AI-3**.

entry	Cat	x mol%	t (h)	yield of 2a	2a : 3a	TON	TOF (h ⁻¹)
1	AI-2	5 mol%	20 h	94%	18 : 1	19	1 h ⁻¹
2	AI-2	2 mol%	20 h	61%	14 : 1	31	1.6 h ⁻¹
3	AI-2	1 mol%	20 h	45%	18 : 1	45	2.3 h ⁻¹
4	AI-3	5 mol%	20 h	93%	14 : 1	19	1 h ⁻¹
5	AI-3	1 mol%	20 h	88%	>20 : 1	88	4.4 h ⁻¹
6	AI-3	0.5 mol%	48 h	89%	18 : 1	178	3.7 h ⁻¹

4. The inclusion of a catalytic cycle diagram in the manuscript is recommended to graphically elucidate the mechanistic pathway. This visual representation would significantly enhance conceptual clarity for readers assessing the proposed reaction mechanism.

Response: We thank the reviewer for this valuable suggestion, which enables a clearer and more concise presentation of the proposed mechanism. As shown in Figure 6, the catalytic cycle begins with hydrogen transfer from the bridging quinoline to the external quinoline, which generates species **B** through aromatization of the bridging quinoline and concurrent dearomatization of the external quinoline. Subsequent hydrogen migration from the aluminum center to the bridging quinoline forms species **C**, followed by hydrogenation of the Al-N bond with ammonia borane via a six-membered transition state to form **F**. Finally, ligand exchange with quinoline releases the 1,2-dihydrogenated product and regenerates the catalytic species **A**. The proposed catalytic cycle has been included into the revised manuscript.

Figure 6. The proposed catalytic cycle.

5. The substrate scope results indicate that while 2-methylquinoline did not react, product 2I was obtained in 99% yield. The observed inactivity attributed to steric effects requires further validation.

Testing 2,4-disubstituted quinolines and supporting the findings with comparative DFT calculations would be beneficial.

Response: Thank you to the reviewer for this thoughtful question.

Calculations show that the presence of a methyl substituent of the C2 position is expected to have a dramatic effect kinetics of the reaction of the respective quinoline with the dialumene. The barrier is higher by 12.7 kcal mol⁻¹. With $\Delta G^\ddagger = 35.5$ kcal mol⁻¹ the reaction with 2-methylquinoline is not expected to occur (Figure S54).

Figure S54. Calculated reaction pathway for the reaction of **Al-1** with 2-Me quinoline.

As suggested, we evaluated the reactivity of 2,4-dimethylquinoline under the same conditions. No reaction was observed, even at elevated temperatures, consistent with the behavior of 2-methylquinoline.

Regarding the contrasting reactivities of 2-methylquinoline (**1h**) and acridine (**1l**), this difference can be partially attributed to the structural nature of the heterocyclic substrates. In the case of polycyclic systems such as acridine (**1l**), the reduction process involves breaking the aromaticity of one of the three fused rings, which is comparatively more favorable than disrupting one of the two rings in quinoline. This is due to the inherent difficulties associated with breaking the

aromaticity of N-heterocycles. That's also why pyridine is much more resistant to reduction than quinoline. In addition, the stoichiometric reaction of **Al-1** with **1I** formed an ill-defined mixture, implying that the catalytic hydrogenation of **1I** may undergo a completely different mechanism.

6. To further validate the existence of π - π stacking interactions and their significance in stabilizing intermediate **A**, non-covalent interaction (NCI) plots or energy decomposition analysis (EDA) might be helpful.

Response:

To demonstrate the non-covalent interaction between the quinoline moieties, we use the reduced density gradient (RDG) function (Figure S49). The isosurface (iso = 0.5) represents the weak interaction regions between aromatic moieties.

Figure S49. Isosurface of the reduced density gradient (RDG) function of **A**, at PBE0-D4/def2-TZVP//r²SCAN-3c level of theory.

The quantum theory of atom in molecules (QTAIM) molecular graph (Figure S50) also displays interactions between the quinoline moieties, showing seven (3, -1) critical points.

Figure S50. QTAIM molecular graph of **A** showing the bond paths and bond critical points, at PBE0-D4/def2-TZVP//r²SCAN-3c level of theory.

These calculations along with the geometrical features support the assignment of the existing π - π stacking interactions between the quinoline moieties in **A**.

Reviewer #1 (Remarks to the Author):

The authors have addressed the main concerns and improved the manuscript based on the suggestions. It may be accepted for Nature Communications.

Response: We are grateful for the reviewer's valuable input on our work.

Reviewer #2 (Remarks to the Author):

I am satisfied that the authors have made significant efforts to improve the manuscript based on feedback from all three reviewers. It is a shame that the proposed active catalyst A could not be detected – though this is perhaps not surprising in a very active system.

I have two minor comments, but I don't feel this hinders publication.

1. The results of the more detailed catalytic investigations are interesting. I do feel that these results could be slightly better integrated in terms of the narrative, aiding the understanding of the system.

2. The proposed formation of A from Al-1/Al-3 is quite longwinded – I would encourage the author to continue investigating the solution behaviour of Al-3 to provide experimental support for the formation of A.

I noted a few minor typos and odd abbreviations, but these can be addressed in the edit.

Otherwise, I would consider this revised manuscript of suitable standard and impact for publication on Nature Communications.

Response: We thank the reviewer for the valuable suggestions. We agree that identifying the catalytic species is crucial for understanding the catalytic system. As recommended, we conducted additional experiments to gain insights into the potential intermediates. We attempted to use UV-Vis spectroscopy to detect species generated in situ from the reaction of Al-3 with quinoline or ammonia borane. However, no characteristic peaks were observed in the UV-Vis spectra. Despite this, we are continuing to investigate the chemical behavior of Al-3 to gain a deeper understanding of the reaction mechanism. A sentence has been added to the end of the manuscript: "Further studies aimed at elucidating the reaction mechanism and expanding aluminum-mediated catalytic transformations are currently underway."

Reviewer #3 (Remarks to the Author):

The authors have taken all comments seriously and prepared a rigorous revised version of the manuscript. I am happy to recommend acceptance in its current form.

Response: We thank this reviewer for the recommendation.